# Isoflavone Consumption and Risk of Breast Cancer: An Updated Systematic Review with Meta-Analysis of Observational Studies

**DOI:** 10.3390/nu15102402

**Published:** 2023-05-21

**Authors:** Jining Yang, Hui Shen, Mantian Mi, Yu Qin

**Affiliations:** 1Research Center for Nutrition and Food Safety, Institute of Military Preventive Medicine, Third Military Medical University (Army Medical University), Chongqing 400038, China; 2Chongqing Medical Nutrition Research Center, Chongqing 400038, China

**Keywords:** isoflavone, breast cancer, meta-analysis, dose–response

## Abstract

Rationale: Epidemiological studies that focus on the relationship between dietary isoflavone intake and the risk of breast cancer still lead to inconsistent conclusions. Herein, we conducted a meta-analysis of the latest studies to explore this issue. Method: We performed a systematic search using Web of Science, PubMed, and Embase from inception to August 2021. The robust error meta-regression (REMR) model and generalized least squares trend (GLST) model were used to establish dose–response relationships between isoflavones and breast cancer risk. Results: Seven cohort studies and 17 case-control studies were included in the meta-analysis, and the summary OR for breast cancer was 0.71 (95% CI 0.72–0.81) when comparing the highest to the lowest isoflavone intake. A subgroup analysis further showed that neither menopausal status nor ER status has a significant influence on the association between isoflavone intake and breast cancer risk, while the isoflavone intake doses and study design does. When the isoflavones exposure was less than 10 mg/day, no effects on breast cancer risk were detected. The inverse association was significant in the case-control studies but not in the cohort studies. In the dose–response meta-analysis of the cohort studies, we observed an inverse association between isoflavone intake and breast cancer: a 10 mg/day increase in isoflavone intake was related to reductions of 6.8% (OR = 0.932, 95% CI 0.90–0.96) and 3.2% (OR = 0.968, 95% CI 0.94–0.99) in breast cancer risk when using REMR and GLST, respectively. In the dose–response meta-analysis of the case-control studies, the inverse association for every 10 mg/day isoflavone intake was associated with breast cancer risk reductions by 11.7%. Conclusion: present evidence demonstrated that taking in dietary isoflavone is helpful in reducing the breast cancer risk.

## 1. Introduction

Breast cancer is the most commonly diagnosed cancer among women worldwide, with an estimated 2.3 million new cases in 2020, and the leading cause of cancer death, with 685,000 deaths in 2020 [1]. The number of new cases is expected to reach 4.4 million in 2070 [2], with patterns and trends varying in different countries. The incidence is generally higher in Western countries than in Asia, but it has quickly increased among Asian women, which has been attributed to the westernization of lifestyle [3], suggesting that in addition to genetic factors, lifestyle factors may contribute to the etiology of breast cancer. The role of estrogen in breast cancers has become increasingly evident. Soy, a traditional and popular food in Asian countries, contains isoflavone, which resembles 17β-estradiol and, thus, has the ability to bind to and activate estrogen receptors (ERs) in breast cancer [4]. It has been suggested that isoflavones play a role in reducing breast cancer risk by reducing the production of estrogen and reactive oxygen species and inhibiting cell proliferation, as per mechanistic studies [4]. However, conclusions on the relationship between isoflavone intake and breast cancer risk in epidemiological studies are still inconsistent [5,6,7,8].

Several meta-analyses have been performed in the past trying to draw a conclusion on the relationship between isoflavone intake and breast cancer risk [9,10,11]. However, their conclusions remain inconsistent due to the differences in their study inclusion criteria, among others. In addition, a considerable number of observational studies have been carried out since our team’s previous systematic review on this topic [12], which added new evidence on the relationship between isoflavone intake and breast cancer risk. Therefore, an updated meta-analysis is required to re-evaluate the relationship between isoflavone intake and breast cancer risk, including subgroup analyses and dose–response analyses.

In this systematic review, a comprehensive meta-analysis was performed to evaluate the probability association between dietary intake levels of isoflavones and breast cancer risk. Subgroup analyses were also carried out to assess the varieties of breast cancer risk among baseline characteristics such as menopausal status and ethnic groups. In addition, to provide a more clarified picture of the relationship between isoflavone intake levels and breast cancer, we conducted a dose–response meta-analysis to interpret our results.

## 2. Materials and Methods

### 2.1. Protocol and Guidance

This study is a systematic review and meta-analysis of isoflavone consumption and the risk of breast cancer in female humans. The recommendations of the Preferred Reporting Items for Systematic Reviews and Meta-Analyses (PRISMA) were followed. The protocol of this study was registered in the International Prospective Registry for Systematic Reviews (PROSPERO) in advance (CRD42021289115).

### 2.2. Inclusion Criteria

This review included observational studies (cohort or case control) undertaken in the adult female population that reported on the association between isoflavone consumption and the risk of breast cancer from January 2000 to August 2021, with a detailed description of the isoflavone intake dose, as well as the estimated effects in the form of a hazard ratio (HR) or odds ratio (OR) with a 95% confidence interval (95% CI). If data from the same cohort were used for different publications, we only included the latest published study to avoid replicated data. Due to the low absolute risk of breast cancer, all measures of relationships, including RR, IRR, and HR, are more likely equal to the estimates of the OR [13], which was used as the study’s outcome.

### 2.3. Exclusion Criteria

Studies conducted in vitro or with animals, reviews, notes, reports, short surveys, conference letters, and case reports were excluded. Studies that used urine or serum markers instead of pathological methods for the diagnosis of breast cancer were excluded. Studies without normalization or without analysis of 95% CIs were excluded as well.

### 2.4. Search Strategy

The PICO strategy (Population, Intervention/Exposure, Comparison, Outcomes) was used to determine an article’s eligibility (Appendix A). A systematic search was carried out using the following public databases: Web of Science, PubMed, and Embase, from the database’s inception to 1 August 2021. Neither data nor language restrictions were applied. Two groups of keywords were utilized to search for articles through the Medical Subject Headings (MeSH). One group used to search for isoflavone were “isoflavones”, “genistein”, “daidzein”, “glycitein”, “biochanin”, “formononetin”, “soy”, “red clover”, “tofu”, and “soya”. The other group for breast cancer was “breast cancer”, “breast carcinoma”, “breast neoplasm”, and “breast tumor”. Details concerning the search strategies for each database are presented in Appendix A.

### 2.5. Study Selection and Data Collection

After automatically removing the duplicates using Endnote X9, all titles and abstracts considered for potential inclusion were screened by two independent authors (J. N. Y. and H. S). Multiple publications by the same author, cohort, or institution were rigorously reviewed, and the studies with the most cases were included to avoid incorporating duplicates.

Two authors (J. N. Y. and H. S.) applied a standardized method to extract data independently. The following parameters were extracted: authors’ information, publication year, study design, country, study period, number of cases and subjects, age, diagnosis of breast cancer, isoflavone exposure range, exposure assessment, risk estimates with corresponding 95% CIs, and adjustment for potentially confounding factors.

Any disagreements during the study selection or data collection were resolved by discussion until a consensus was reached or by consulting another author (Y. Q.).

### 2.6. Quality Assessment of Evidence

Using the Newcastle–Ottawa Scale (NOS), two authors (J. N. Y. and H. S.) independently assessed the quality of each study [14]. Any disagreements were resolved by discussion until a consensus was reached or by consulting another author (Y. Q.).

### 2.7. Statistical Analysis

The statistical analyses were conducted using STATA version 16 (Stata Corp). *p*-Values were considered significant at the level of *p* < 0.05.

#### 2.7.1. Meta-Analysis

The ORs and their 95% CIs were used to report the results. We performed a random-effects meta-analysis to estimate the pooled effect size for the highest in comparison with the lowest intakes of isoflavones. To assess the heterogeneity among studies, we calculated the Cochran Q and the I^2^ statistics [15]. For the Q statistic, *p* < 0.1 was considered significant. For the I^2^ statistic, a low level of heterogeneity was considered when I^2^ was less than 40%, a medium level for 40% < I^2^ < 60%, and a high level for I^2^ > 60%. A multivariate meta-regression analysis was conducted to identify the source of the heterogeneity.

#### 2.7.2. Subgroup Analysis

For additional insight, subgroup analyses were conducted based on the study design (cohort study or case-control study), population (Asian or non-Asian), publication year (2001–2010 or 2010–2021), study quality (NOS score ≥ 7 or NOS score < 7), menopausal status (premenopausal or postmenopausal), isoflavone intake doses (≥10 mg or <10 mg), and estrogen receptor (ER) status (positive or negative).

#### 2.7.3. Dose–Response Analysis

We selected the mean values of the isoflavone intake doses in each category to represent the average intake level. For studies that reported a range of isoflavone intakes, the midpoint of the upper and lower boundaries was regarded as the intake dose of each category. When the highest categories were open-ended, the dose was considered 1.2-fold that of the highest boundary [16].

The dose–response meta-analysis was performed in the cohort studies and case-control studies. In this analysis, we utilized a one-stage, robust error meta-regression (REMR) model [17] and a two-stage, generalized least squares trend (GLST) model [18] to estimate the potential curvilinear association between isoflavones and the risk of breast cancer. The GLST model’s method requires that the studies report ≥3 exposure categories, which was not specified in the REMR model. In both of the two methods, we modeled the dose using a restricted cubic spline model with 3 knots at 5%, 50%, and 95% of the distribution. In order to test the nonlinearity, the Wald test was used, with the null hypothesis that the coefficient of the second spline was zero. The GLST model and REMR model were used to investigate the linear dose–response association between a 10 mg/day increment in intake of isoflavones and the risk of breast cancer. Statistical analyses were performed using STATA version 16.0 (Stata Corp). *p*-Values were considered significant at the level of *p* < 0.05.

#### 2.7.4. Risk of Publication Bias and Sensitivity Analysis

Sensitivity analyses were performed to identify whether the overall estimates were dependent on the effect size from a single study by excluding each study in turn [19]. Publication bias was evaluated using Begg’s funnel plot and Egger’s regression test [20]. The trim-and-fill method was performed for further analysis, if the Egger’s regression test was significant [21].

## 3. Results

### 3.1. Characteristics of the Studies

As shown in Figure 1, 5577 studies were identified based on our search strategy. After removing duplicate records and records marked as ineligible by automation tools, which screened out studies with “cell”, “mouse”, “rat”, etc., in the title or abstract, 2337 studies remained. After assessing the titles and abstracts of these studies, 2127 records were subsequently excluded based on our inclusion and exclusion criteria. After assessing the full text of the remaining 210 studies, 94 records were excluded because of the unavailability of the isoflavone intake dose data. After a thorough final review of the 116 remaining studies, 92 studies were excluded because they were unrelated to our topic (n = 66) and or because of data duplication (n = 26). There were 24 studies [5,6,7,8,9,22,23,24,25,26,27,28,29,30,31,32,33,34,35,36,37,38,39,40] that met our inclusion criteria and were included in this meta-analysis.

The overall quality of the included studies was relatively high. The average NOS score of the 24 included studies was 7.6, ranging from 6 to 9. Among these studies, eight achieved a score of 7, nine achieved a score of 8, and four achieved a score of 9. Detailed outcomes of the quality assessment for each study are presented in Appendix A.

A summary of the characteristics of the included studies is provided in Table 1 and described below. Of the 24 included studies, 17 were case-control studies [5,6,22,23,24,25,26,27,28,29,30,31,32,33,34,35,36] and 7 were cohort studies [7,8,9,37,38,39,40]. The selected studies were published between 2001 and 2021 and were carried out in Asia [6,8,9,26,27,28,29,30,31,33,34,35,36,37,38,40], America [5,22,23,27,32,39], and Europe [7,24,25]. The study-specific, maximally adjusted ORs or HRs were extracted and pooled for the meta-analysis to evaluate the association of isoflavone and the risk of breast cancer in a total of 902,438 females. The verification of breast cancer in these studies was based on either a cancer registry record or a histological diagnosis. The exposure assessment of all included studies was based on a food frequency questionnaire (FFQ) via either face-to-face interviews or self-administrative questionnaires.

### 3.2. Meta-Analysis of Isoflavone Consumption and Risk of Breast Cancer

From each study and all studies combined, the estimated ORs for the highest versus lowest levels of isoflavone dietary intake are shown in Figure 2. A protective effect of isoflavone dietary intake on breast cancer risk (OR 0.71, 95% CI 0.62–0.81) were performed, but unneglectable heterogeneity existed (*p* < 0.001 for heterogeneity, I^2^ = 82.6%).

Therefore, a meta-regression was conducted to determine the possible sources of heterogeneity. In the meta-regression analysis, we explored the study design (cohort vs. case-control study), population (Asian vs. non-Asian), publication year, study quality, menopausal status (premenopausal vs. postmenopausal), and isoflavone intake dose and case number. As a result of the meta-regression, study design (*p* = 0.017), population (*p* = 0.009), and isoflavone intake dose (*p* = 0.038) were identified as potential sources of heterogeneity.

### 3.3. Subgroup Analysis of Isoflavone Consumption and the Risk of Breast Cancer

We performed a subgroup analysis according to the study design (cohort vs. case-control study), population (Asian vs. non-Asian), publication year (2001–2010 vs. 2010–2021), study quality (NOS score ≤ 7 vs. NOS score > 7), menopausal status (premenopausal vs. postmenopausal), isoflavone intake doses ≥ 10 mg vs. <10 mg), and ERs status (positive vs. negative). The pooled ORs are presented in Table 2, and detailed forest plots are provided in Appendix A. As the results show, the subgroup analysis based on isoflavone intake dose (*p* for interaction <0.001), population (*p* for interaction <0.001), and study design (*p* for interaction <0.001) could explain the between-study heterogeneity.

A statistically significant protective effect of isoflavone intake on breast cancer was observed in the case-control studies (OR = 0.62, 95% CI 0.50–0.76), while no such effect was observed in the cohort studies (OR = 0.94, 95% CI 0.86–1.02). In addition, the pooled OR showed an inverse relationship between isoflavone intake and breast cancer in Asian women (OR = 0.62, 95% CI 0.52–0.74), while the relationship did not exist in n7uon-Asian women (OR = 0.97, 95% CI 0.88–1.06). When the highest isoflavone intake was lower than 10 mg/d, the negative relationship between isoflavone intake and breast cancer disappeared (OR = 1.01, 95% CI 0.94–1.08), whereas OR = 0.63, 95% CI 0.53–0.75, when the highest isoflavone intake was above 10 mg/d. However, a statistically significant difference in the protective effect of isoflavone intake on breast cancer was observed regardless of whether the women were pre- or postmenopausal and regardless of whether they were ER positive or negative.

### 3.4. Dose–Response Meta-Analysis of Isoflavone Consumption and the Risk of Breast Cancer

Next, we assessed the dose–response relationship between isoflavone intake and breast cancer risk using both REMR and GLST methods for the case-control studies and cohort studies, respectively.

The cohort studies included seven studies both in the GLST model and the REMR model. When using the REMR method, the *p*-value for the nonlinear association was 0.0081. Therefore, the curvilinear dose–response REMR was used. We found that a 10 mg/day increase in isoflavone intake was associated with a 6.8% lower risk of breast cancer (OR = 0.932, 95% CI 0.90–0.96, *p* = 0.002) (Figure 3A). When using the GLST method, the *p*-value for the nonlinear association was 0.1141. Thus, the linear dose–response GLST was used. The pooled OR for breast cancer risk at a 10 mg/day increment in isoflavone dietary intake was 0.968 (95% CI 0.94–0.99, *p* = 0.009), which means there was a 3.2% decrease in the risk of breast cancer for an increase of 10 mg isoflavone intake per day (Figure 3B). It can be observed that the breast cancer risk is significantly reduced with an isoflavone intake of approximately 15 mg/day when using the REMR method.

For the case-control studies, we finally involved 12 and 17 studies in the GLST model and REMR model, respectively. When using the REMR method (Figure 3C), we found a curvilinear association between isoflavone intake and breast cancer (*p* nonlinearity = 0.0129). With each 10 mg/day increment in isoflavone intake, the pooled risk of breast cancer was reduced by 11.7% (OR = 0.88, 95% CI 0.85–0.91, *p* < 0.001). When using the GLST method, we found a curvilinear association between isoflavone intake and breast cancer (*p* nonlinearity = 0.0002), an increase of 10 mg/day isoflavones intake was associated with a 19.3% lower risk of breast cancer (OR = 0.81, 95% CI 0.78–0.84, *p* < 0.001) (Figure 3D). The decrease in the breast cancer risk was slightly decelerated when the isoflavone intake was >20 mg/day, but no significant slowdown was observed using both methods.

### 3.5. Risk of Publication Bias

The publication biases were evaluated using Begg’s test and Egger’s test. The shape of the funnel plots showed asymmetry (Appendix A, *p* = 0.001), and the Egger’s test found virtual publication bias (Appendix A, *p* < 0.001). However, the trim-and-fill method failed to identify any potentially missing studies (Appendix A), indicating the publication bias did not affect the results.

### 3.6. Sensitivity Analysis

By excluding individual studies at a clip, the contribution of individual studies on the overall results was assessed by testing the presence of noticeable changes in the overall result (Appendix A). In general, a few studies contributed intensive but insignificant weight to the overall results.

## 4. Discussion

In this observational-based meta-analysis, our results showed that dietary isoflavone intake has a negative correlation with breast cancer risk, suggesting a potential protective effect of isoflavone on breast cancer. However, we also noticed the non-negligible heterogeneity in the overall results. Using a meta-regression analysis, we found that the study design, isoflavone intake dose, and population might be the major sources of the heterogeneity. Regarding these concerned factors, we further conducted a subgroup analysis.

When we stratified by menopause status, estrogen receptor, publication year, and NOS score, the negative correlation was still significant, indicating that these factors might not be the key sources of the heterogeneity. Interestingly, the association was almost the same regarding menopause status (OR 0.76, 95% CI 0.63–0.92 in premenopausal vs. 0.75, 95% CI 0.62–0.90 in postmenopausal) and estrogen receptor (0.77, 95% CI 0.62–0.95 in ER+ vs. 0.77, 95% CI 0.52–1.15 in ER-), suggesting that menopause status and estrogen receptor had little effect on the correlation between isoflavone and breast cancer risk. Previous laboratory studies show that the sexual hormone was an important risk factor for breast cancer, and estrogen therapy is effective and validated [41]. Mechanistically, isoflavone is an estrogen-like compound that can bind and activate estrogen receptors (ERs) in breast cancer and, thus, was indicated to be effective in curing breast cancer [42]. However, our meta-analysis result showed that the up-to-date epidemiological evidence could not support the estrogen-like effect on breast cancer. One interesting hypothesis of the preventive effects of isoflavone on breast cancer is its inhibition of cancer initiation, which occurs at an early age when the cells are in good shape [43,44]. This means that the starting exposure time of isoflavone is important for the prevention of breast cancer. However, a limited number of epidemical studies have been published that support the hypothesis [23,32,45]. A case-control study in Asian American females indicated that the high consumption of isoflavone during adolescence was related to a reduced risk of breast cancer, even though the consumption was low during adult life [23]. Thanos et al. proposed that higher exposure to isoflavones during adolescence exhibited a decreased risk of breast cancer among non-Asian women [45], and a subsequent study by this team showed that the decreased risk may be relative to the ER and PR status [32]. Further, a recent large prospective cohort study that included more than 11,000 women aged above 50 reported that isoflavone supplements had a significant positive association with ER-negative breast cancer, although there was a significant inverse association with ER-positive breast cancer [46], which is inconsistent with our meta-analysis results that included all adult female data, further suggesting that the intake of isoflavone at an early age might greatly benefit breast cancer.

The subgroup analysis of the publication year and NOS score showed that the negative correlation was stronger in studies with an NOS score above seven than below seven, and it was stronger in studies published before 2010 than after 2010. The results suggest that some characteristics of the study may influence the risk estimate, although the combined ORs were still significant. One explanation could be the uncertainties in the isoflavone intake assessment. Soy, an isoflavone-rich food, is more often used as an ingredient in transformed food nowadays than in earlier times [47,48]. This ingredient significantly increases the exposure to isoflavones even in “non-soy eaters”, which could probably underestimate the isoflavone intake dose, thus increasing the heterogeneity in recent studies.

The subgroup analysis of the study design showed that the negative correlation between dietary isoflavone intake and breast cancer risk was only significant in case-control studies but not in cohort studies. Zhao et al. [10] also drew a similar conclusion in their recent meta-analysis. They performed a meta-analysis on cohort studies of dietary isoflavones intake and breast cancer risk, which showed no significant association between isoflavones intake and the risk of breast cancer regardless of high isoflavone intake (RR 0.99, 95% CI 0.91–1.09) or low intake (RR 0.99, 95% CI 0.92–1.05), although the included criteria were slightly different with our present study. In addition, other factors, including isoflavone intake dose and population, also contributed to the heterogeneity of our meta-analysis. Different populations have different eating habits. Specifically, Asians consume much more isoflavone-rich soy foods in their daily life, such as tofu in China and miso-soup in Japan. The studies on Asian populations and relatively high isoflavone intake doses are mostly consistent with each other. The subgroup analysis of the isoflavone intake dose showed that only the relatively high isoflavone intake (>10 mg/day) was negatively correlated with breast cancer, indicating that the protective effects of isoflavone on breast cancer might only occur when people take in enough isoflavone each day. Furthermore, the negative correlation between isoflavone intake and breast cancer risk was only observed in Asian adult women but not in non-Asian adult women. This phenomenon could partly be attributed to the different isoflavone intake doses due to the different eating habits, since Asian populations usually consume more isoflavone-rich food compared with non-Asian populations. As the previous studies report, the isoflavone intake of Asian populations ranges from 15 to 60 mg/day [49,50], which is merely <3 mg/day in non-Asian populations [51,52]. However, whether gene polymorphisms played a role remains to be explored. A similar trend in isoflavone intake has been demonstrated in the studies involved in the previous meta-analysis [12].

Considering the vital role of the isoflavone intake dose in breast cancer risk, we further performed a dose–response meta-analysis of isoflavone consumption and the risk of breast cancer. As the study design is another important factor that contributes to the heterogeneity, we performed a dose–response meta-analysis within the case-control studies and cohort studies, respectively. The GLST approach used in this article is a commonly used two-stage dose–response meta-analyses method. Using this method, we found a 3.2% decrease in the risk of breast cancer for an increase of 10 mg isoflavone intake per day in the cohort studies, which is consistent with Wei et al.’s results using the GLST method (HR 97% for every 10 mg isoflavone intake, 95% CI 0.95–0.99) [9]. Using the same method, a 19.3% decrease in the risk of breast cancer for an increase of 10 mg isoflavone intake per day was found in the case-control studies, which is much higher than the results in the cohort studies. However, the dose–response analysis of isoflavone intake and breast cancer within the case-control studies was not found in recent studies. Wang et al. reported a 10% (95% CI 7%–13%) decrease in the risk of breast cancer for an increase of 10 g tofu (an isoflavone-rich food) intake per day, within five case-control studies, using the same method. If converting tofu intake to isoflavone intake based on the median (237 µg isoflavone in 1 g tofu) in the previous study [53], the dose–response result of Wang et al. should be a 42.2% decrease in the risk of breast cancer for an increase of 10 mg isoflavone intake per day, which is also a substantially high number. The second method used in our study was RMER, a one-stage method for dose–response meta-analysis, which was recommended in a recent study [17]. Using this method, the decreased rate of breast cancer for every 10 mg of isoflavones was 6.8% and 11.7%, respectively, within the cohort studies and case-control studies. However, no comparable data were reported.

Although the association between isoflavone and breast cancer is dose-dependently inverse, the potentially detrimental effects of isoflavone at very high doses cannot be neglected. The leading concern is the potential detrimental effects on postmenopausal women, since high concentrations of estrogen could lead to endometrial carcinoma in aged women [54]. One RCT study reported that no significant adverse effects were observed in 224 postmenopausal women the United States who received supplementation with soy isoflavone at 120 mg/day for 3 years [55]. Another RCT study reported that six women developed endometrial hyperplasia but no endometrial cancer in 319 Italian postmenopausal women supplemented with 150 mg/day of soy isoflavones for 5 years [56]. More recently, no endometrial thickness hyperplasia or histopathological changes were observed in 399 postmenopausal women in Taiwan who were supplemented with 300 mg/day of soy isoflavones for 2 years [57] and 350 postmenopausal women in the United States supplemented with 154 mg/day of soy isoflavones for three years [58]. However, other potential hazards to the reproduction of a high dose of isoflavone should be taken into consideration. Some recent observational studies report potential detrimental effects on reproduction of isoflavone with high doses in Western women [59,60], which should also be taken into consideration. However, up-to-date evidence supports that a dose below 50 mg/day may not induce reproductive impairment.

There are several limitations in this study that should be noticed. First, in order to have a full view of the correlation between dietary isoflavone consumption and breast cancer risk, we included as much clinical evidence as possible according to our criteria, leading to the inevitable heterogeneity among study designs. In addition, quite a few of the studies included were case-control studies (17 of 24 studies). Therefore, recall and selection biases should be noticed. However, we tried to limit these biases by only keeping the cohort study data when a parallel case-control study was taken using the same cohort of this cohort study. Second, the isoflavone intake might be underestimated in Western countries, since soy is used as an ingredient in many transformed dishes, which might lead to a bias in the subgroup analysis stratified by intake dose. Third, in each study included in the present meta-analysis, different methods of assessing the intake dose of isoflavone were employed. For example, self-administered questionnaires, mail survey questionnaires, and multitype food-frequency questionnaires were used in these studies, which could potentially lead to bias. Fourth, the adjusted factors were different in each included study and, thus, might bias the association between dietary isoflavone intake and breast cancer risk. Fifth, genistein, daidzein, and glycitein are the major food-derived isoflavones, with different distributions in different food legumes and soy products [61]. In addition, there are certain differences in the metabolites and biological effects among them. For example, the bioavailability is higher in genistein [62], but the equol, the metabolite of daidzein, exerts significantly greater antioxidant activity and estrogenic activity on binding to the ER receptor compared with daidzein, thus leading to different biological effects [63]. However, most of the observational studies did not distinguish them in the FFQ survey, and only the total isoflavones intakes were calculated, which might lead to evitable heterogeneity among countries with different food preferences. As a result, the effect of various isoflavones on breast cancer risk requires further investigations.

In conclusion, evidence from observational studies suggest a dose–response inverse association between dietary isoflavone intake and breast cancer risk. However, heterogeneity and biases in the present studies require further well-designed prospective cohort and RCT studies to confirm our findings.

## 5. Conclusions

This updated meta-analysis showed that dietary isoflavone intake has a protective effect on breast cancer risk, with a significant dose–response correlation, regardless of premenopausal or postmenopausal conditions. Future prospective cohort studies or RCTs are also needed to determine the causality of this relationship.

## Figures and Tables

**Figure 1 nutrients-15-02402-f001:**
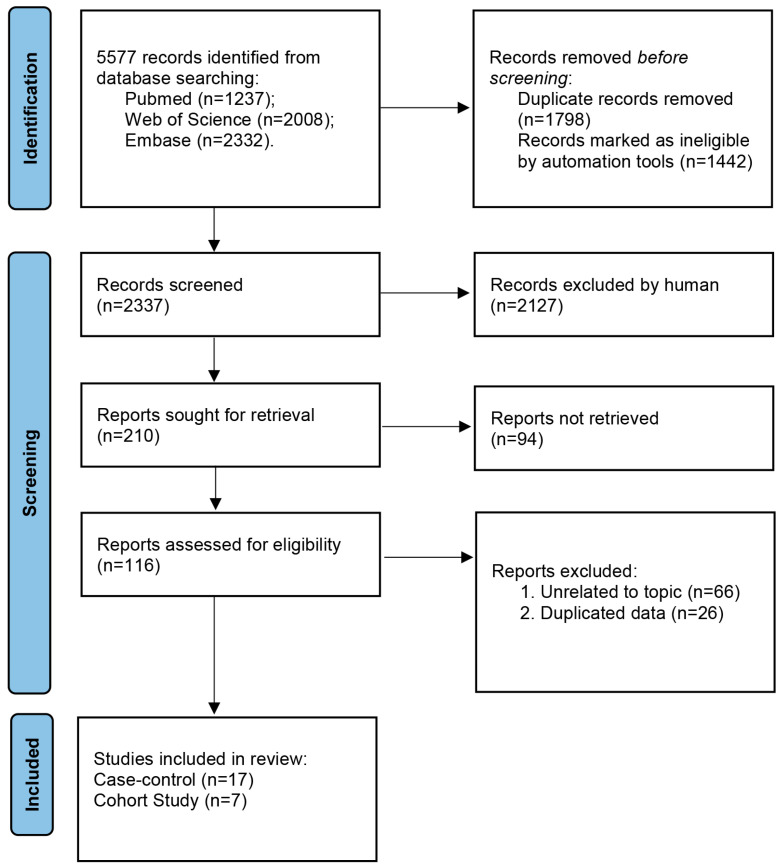
Flowchart of the study’s screening process.

**Figure 2 nutrients-15-02402-f002:**
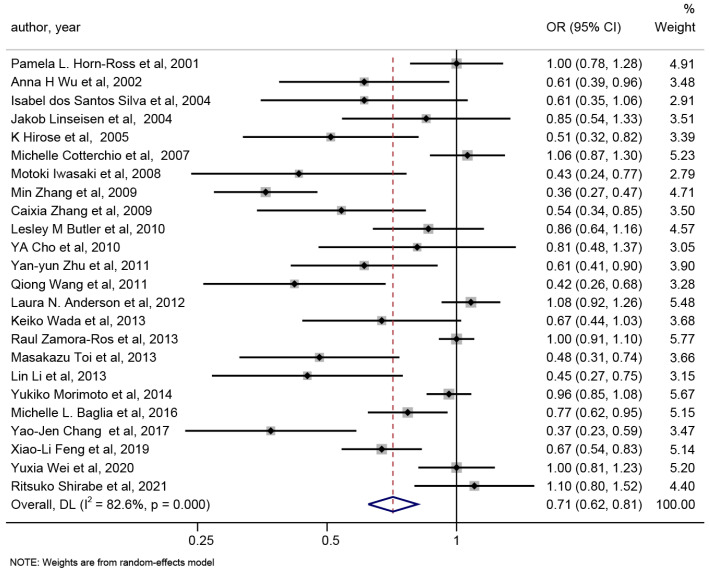
Forest plot of the association between isoflavone intake and risk of breast cancer (highest versus lowest category meta-analysis). The size of the box representing the point estimate for each study in the forest plot is in proportion to the contribution of that study’s weight estimate to the summary estimate. The horizontal lines indicate the 95% CIs. The diamond denotes the pooled odds ratio. The dotted line represents the average of pooled odds ratio [5,6,7,8,9,22,23,24,25,26,27,28,29,30,31,32,33,34,35,36,37,38,39,40].

**Figure 3 nutrients-15-02402-f003:**
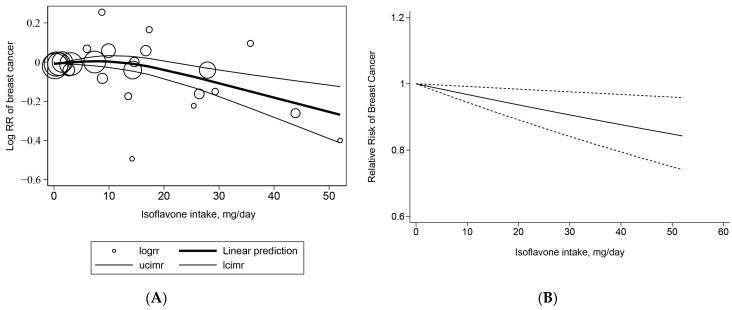
Dose–response meta-analysis of the association between isoflavone dietary intake and breast cancer risk: (**A**) cohort studies, estimated using the REMR model; (**B**) cohort studies, estimated using the GLST model; (**C**) case-control studies, estimated using the REMR model; (**D**) case-control studies, estimated using the GLST model.

**Table 1 nutrients-15-02402-t001:** Characteristics of studies included in the meta-analysis.

Author	Publication Year	Study Design	Population	Study Period	Case/Subjects	Verification of Breast Cancer	Exposure Range (mg/day)	Variables of Adjustment	Exposure Assessment
Keiko Wada	2013	Cohort	Asian	1992–2008	172/15,607	Cancer registry record	<19.9 vs. >67.4	2, 3, 5, 6, 7, 8, 11, 15, 22, 28	FFQ
Lesley M Butler	2010	Cohort	Asian	1993–2005	629/34,028	Histologically confirmed	<4.6 vs. >33.9	2, 6, 7, 10, 12, 22, 42, 43	FFQ
Michelle L. Baglia	2016	Cohort	Asian	1996–2011	1034/70,578	Cancer registry record	<11.1 vs. >55	2, 5, 7, 8, 10, 12, 22, 28, 44	FFQ
Raul Zamora-Ros	2013	Cohort	Non-Asian	1992–2010	11,576/334,850	Histologically confirmed	<0.22 vs. >1.36	2, 3, 5, 7, 8, 9, 11, 12, 13, 15, 20, 22, 28, 33	FFQ
Ritsuko Shirabe	2021	Cohort	Asian	1990–2013	825/46,714	Histologically confirmed	<9.1 vs. >44.8	2, 3, 6, 10, 11, 12, 13, 14, 15, 18, 19, 20, 22, 23, 28, 45	FFQ
Yukiko Morimoto	2014	Cohort	Non-Asian	1993–2007	4769/84,550	Cancer registry record	<3.2 vs. >20.3	1, 2, 3, 4, 5, 6, 7, 8, 9, 10, 11, 12, 13, 14, 15, 16	FFQ
Yuxia Wei	2020	Cohort	Asian	2004–2016	2289/300,852	Cancer registry record	<4.5 vs. >19.1	2, 3, 6, 7, 8, 9, 12, 13, 15, 17, 18, 19, 20, 28, 30	FFQ
Yan-yun Zhu	2011	Case control	Asian	2008~2011	62/108	Histologically confirmed	<7.56 vs. >28.83	10, 13, 15, 21, 22, 30, 31	FFQ
Min Zhang	2009	Case control	Asian	2004–2005	158/410	Histologically confirmed	<7.78 vs. >25.40	2, 3, 7, 8, 9, 10, 12, 13, 15, 22, 23, 24, 28, 31	FFQ
Caixia Zhang	2009	Case control	Asian	2007–2008	140/249	Histologically confirmed	<3.26 vs. >16.89	2, 3, 10, 12, 13, 21, 28, 30	FFQ
Anna H Wu	2002	Case control	Non-Asian	1995–1998	130/278	Cancer registry record	<1.79 mg/1000 kcal vs. >12.68 mg/1000 kcal	2, 3, 7, 8, 10, 11, 13, 15, 22, 25, 26, 27, 28	FFQ
Motoki Iwasaki	2008	Case control	Asian	2001–2006	850/1700	Histologically confirmed	<8.7 vs. >71.3	6, 8, 10, 13, 28, 32	FFQ
Pamela L. Horn-Ross	2001	Case control	Non-Asian	1995–1998	292/694	Cancer registry record	<1.048 vs. >2.774	1, 2, 3, 6, 7, 8, 10, 11, 12, 19, 21, 22	FFQ
K Hirose	2005	Case control	Asian and Non-Asian	2001–2002	36/174	Histologically confirmed	<7.61 vs. >11.87	2, 3, 5, 6, 12, 13, 15, 20, 22, 29, 28	FFQ
Isabel dos Santos Silva	2004	Case control	Non-Asian	1995–1999	239/714	Cancer registry record	<0.125 mg/1862 kcal vs. >0.470 mg/1862 kcal	3, 5, 7, 10, 19, 20, 26	FFQ
Michelle Cotterchio	2007	Case control	Non-Asian	2002–2003	3000/6370	Histologically confirmed	<0.082 vs. >1.237	5, 10, 11, 21, 22, 33	FFQ
Y. A. Cho	2010	Case control	Asian	2007–2008	138/257	Histologically confirmed	<8.5 vs. >23.7	2, 3, 6, 7, 8, 10, 11, 12, 13, 15, 22, 28, 34, 35	FFQ
Masakazu Toi	2013	Case control	Asian	2007–2009	101/266	Cancer registry record	<18.76 vs. >43.75	3, 6, 7, 10, 11, 12, 19, 20, 21, 22, 28, 36, 46	FFQ
Jakob Linseisen	2004	Case control	Non-Asian	1992–1995	278/944	Histologically confirmed	<0.1737 vs. >0.4147	2, 6, 7, 10, 12,13, 19	FFQ
Lin Li	2013	Case control	Asian	2009–2010	113/189	Cancer registry record	<12.49 vs. >35.12	6, 7, 8, 9,10, 12, 13, 15, 17, 24, 28, 31, 37, 38	FFQ
Xiao-Li Feng	2019	Case control	Asian	2007–2008	456/843	Histologically confirmed	<3.41 vs. >13.05	2, 7, 9, 15, 17, 22, 28, 31	FFQ
Yao-Jen Chang	2017	Case control	Asian	2010–2013	233/449	Cancer registry record	<22 vs. >22	22, 28	FFQ
Qiong Wang	2011	Case control	Asian	2007–2009	181/299	Histologically confirmed	<9.95 vs. >23.55	2, 5, 9, 12, 13, 19, 39, 40, 41	FFQ
Laura N. Anderson	2012	Case control	Non-Asian	2002–2003	476/1587	Histologically confirmed	<0.122 vs. >0.497	5, 10, 11, 21, 22, 33	FFQ

FFQ, food frequency questionnaire. Variables of adjustment: (1) ethnicity; (2) BMI; (3) age at menarche; (4) age at cohort entry; (5) age at first live birth; (6) parity; (7) education; (8) menopausal status; (9) oral contraceptive use; (10) family history of breast cancer; (11) menopausal hormone use; (12) total energy intake; (13) smoking; (14) diabetes; (15) alcohol; (16) hypertension; (17) household income; (18) standing height; (19) lactation; (20) age at menopause; (21) history of breast disease; (22) age; (23) residential area; (24) tea drinking; (25) birthplace; (26) pregnancy; (27) dark leafy greens during adolescence; (28) physical activity; (29) motives for consultation; (30) vegetables and fruit intake; (31) passive smoking; (32) vitamin supplement use; (33) dietary fiber intake; (34) current use of dietary supplements; (35) occupation; (36) birth weight; (37) household size; (38) BMI 5 year ago; (39) total protein intake; (40) total fat intake; (41) number of abortions; (42) interview year; (43) dialect group; (44) season of recruitment; (45) received mammography; (46) BMI at the approximate age of 20.

**Table 2 nutrients-15-02402-t002:** Subgroup analyses of isoflavone intake and the risk of breast cancer (highest versus lowest category meta-analysis).

Outcome of Interest	No. of Studies	OR (95% CI)	*p* _heterogenity_	*I^2^* (%)	*p* for Interaction
**Study design**					
Case control	17	0.62 (0.50, 0.76)	0.000	83.8%	0.000
Cohort	7	0.94 (0.86, 1.02)	0.178	32.7%	
**Population**					
Non-Asian	8	0.97 (0.88, 1.06)	0.092	43.0%	0.000
Asian	18	0.62 (0.52, 0.74)	0.000	75.6%	
**Publication year**					
Before 2010	11	0.67 (0.52, 0.87)	0.000	81.5%	0.462
After 2010	13	0.75 (0.64, 0.87)	0.000	82.3%	
**Study quality**					
NOS score ≤ 7	10	0.77 (0.63, 0.93)	0.000	77.4%	0.362
NOS score > 7	14	0.68 (0.56, 0.82)	0.000	84.2%	
**Menopausal status**					
Premenopausal	16	0.76 (0.63, 0.92)	0.000	66.6%	0.897
Postmenopausal	16	0.75 (0.62, 0.90)	0.000	78.3%	
**Isoflavone highest intake**					
<10 mg/d	6	1.01 (0.94, 1.08)	0.452	0.0%	0.000
≥10 mg/d	18	0.63 (0.53, 0.75)	0.000	81.4%	
**ER status**					
ER+	7	0.77 (0.62, 0.95)	0.000	75.7%	0.981
ER−	7	0.77 (0.52, 1.15)	0.000	84.1%	

OR, odds ratio; CI, confidence interval; *p*
_heterogenity_**,**
*p* for heterogeneity within subgroups; NOS, Newcastle–Ottawa Scale; ER, estrogen receptor.

## Data Availability

The data presented in this study are available upon request from the corresponding author.

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
