# Peer review of "Isoflavone Consumption and Risk of Breast Cancer: An Updated Systematic Review with Meta-Analysis of Observational Studies"

_nutrients, 2023, doi:10.3390/nu15102402_

Round 1
Reviewer 1 Report
Major comment
The paper provides a significant public health message suggesting that dietary intake of soy isoflavones is safe and potentially protective against breast cancer. As such, the authors need to discuss other aspects. For example, a recent large prospective cohort study (PMID: 30831601) of more than 11,000 women aged 50 reported that isoflavones supplements had a significant positive association with ER negative breast cancer although a significant inverse association with ER positive breast cancer. In addition, the safety of dietary intake of soy isoflavones among those with a history of breast cancer. (3) Although some studies suggest that a very high dietary intake of soy isoflavones is unsafe, the authors concluded that the association is dose-dependently inverse, which might be misleading. (4) Distinction between genistein and daidzein.
Author Response
Please see the attachment.
Response to Reviewer 1 Comments
The paper provides a significant public health message suggesting that dietary intake of soy isoflavones is safe and potentially protective against breast cancer. As such, the authors need to discuss other aspects.
Point 1: For example, a recent large prospective cohort study (PMID: 30831601) of more than 11,000 women aged 50 reported that isoflavones supplements had a significant positive association with ER negative breast cancer although a significant inverse association with ER positive breast cancer.
Response 1: Thanks for your suggestion. As discussed in the above paper, the different effects of isoflavones regarding ER status might be attributed to its complex estrogen-like effects. However, we also noticed another hypothesis that isoflavone’s preventive effect on breast cancer could be an inhibition of cancer initiation. This should occur at an early age while the breast cells are still in good shape. The cohort comprises women aged more than 50 years while our study enrolled all the adult women, which further suggests the hypothesis that exposure to isoflavone in early life is more beneficial in reducing the risk of breast cancer. The study (PMID: 30831601) has been cited and we discussed the results with ours in line 356-374. The contents in the manuscript are as follows.
“One interesting hypothesis of the preventive effects of isoflavone on breast cancer is its inhibition of cancer initiation, which occurs at an early age when the cells are in good shape[1, 2]. It means that the starting exposure time of isoflavone is important for the prevention of breast cancer. However, a limited number of epidemical studies have been published that support the hypothesis [3-5]. A case-control study in Asian American females indicated that the high consumption of isoflavone during adolescence was related to the reduced risk of breast cancer, even though the consumption was low during adult life[3]. Thanos, et al proposed that higher exposure to isoflavones during adolescence exhibited a decreased risk of breast cancer among non-Asian women[5], and the subsequent study of this team showed that the decreased risk may be relative to the ER and PR status[4]. Further, A recent large prospective cohort study which included more than 11,000 women aged above 50 reported that isoflavones supplements had a significant positive association with ER-negative breast cancer although a significant inverse association with ER-positive breast cancer[6], which is inconsistent with our meta-analysis results that have included all adult female data, further suggesting that taking in isoflavone at an early age might benefit a lot to breast cancer.”
Point 2: In addition, the safety of dietary intake of soy isoflavones among those with a history of breast cancer. Although some studies suggest that a very high dietary intake of soy isoflavones is unsafe, the authors concluded that the association is dose-dependently inverse, which might be misleading.
Response 2: We agree to your point that the safety of isoflavone is of great importance in the present study since we concluded a dose-dependently inverse association of isoflavone and breast cancer. Therefore, the safety of dietary intake of soy isoflavones has been comprehensively discussed in the revised manuscript, line 442-461, to reduce the potential misleading. The contents are as follows.
“Although the association between isoflavone and breast cancer is dose-dependently inverse, the potentially detrimental effects of isoflavone at very high doses cannot be neglected. The leading concern is the potential detrimental effects on postmenopausal women since high concentrations of estrogen could lead to endometrial carcinoma in aged women[7]. One RCT study reported no significant adverse effects were observed in 224 United States postmenopausal women supplemented with soy isoflavone 120 mg/day for 3 years[8]. Another RCT study reported 6 women developed endometrial hyperplasia but no endometrial cancer in 319 Italian post-menopausal women supplemented with 150 mg/day soy isoflavones for 5 years[9]. More recently, no endometrial thickness hyperplasia or histopathological changes were observed in 399 postmenopausal women in Taiwan who were supplemented with 300 mg/day soy isoflavones for 2 years[10] and 350 postmenopausal women in the United States supplemented with 154 mg/day soy isoflavones for three years[11]. However, other potential hazards to the reproduction of high dose of isoflavone should be taken into consideration. Some recent observational studies report the potential detrimental effects on the reproduction of isoflavone with high doses in western women [12, 13], which should also be taken into consideration. Anyway, up-to-date evidence support that a dose below 50 mg/day may not induce reproductive impairment.”
Point 3: Distinction between genistein and daidzein.
Response 3: Thank you for the suggestion. The differences between genistein and daidzein may lead to different biological effects, but most observational studies did not address the intake doses of each kind of isoflavones, which might increase the heterogeneity among studies. Relative points have been added to the limitations of our study in line 479-490. The contents are shown as below.
“Genistein, daidzein and glycitein are the major food-derived isoflavones, with different distributions in different food legumes and soy products[14]. Also, there are certain differences in the metabolites and biological effects among them. For example, the bioavailability is higher in genistein[15], but the equol, the metabolite of daidzein, exerts significantly greater antioxidant activity and estrogenic activity on binding to the ER receptor compared with daidzein, thus leading to different biological effects[16]. However, most of the observational studies did not distinguish them in the FFQ survey and only the total isoflavones intakes were calculated, which might lead to evitable heterogeneity among countries with different food preferences. As a result, the effect of various isoflavones on breast cancer risk require further investigations.”
References:
- Lamartiniere, C.A., et al., Genistein suppresses mammary cancer in rats. Carcinogenesis, 1995. 16(11): p. 2833-40.
- Wu, A.H., et al., Adolescent and adult soy intake and risk of breast cancer in Asian-Americans. Carcinogenesis, 2002. 23(9): p. 1491-6.
- Anderson, L.N., et al., Phytoestrogen intake from foods, during adolescence and adulthood, and risk of breast cancer by estrogen and progesterone receptor tumor subgroup among Ontario women. Int J Cancer, 2013. 132(7): p. 1683-92.
- Khan, S.A., et al., Soy isoflavone supplementation for breast cancer risk reduction: a randomized phase II trial. Cancer Prev Res (Phila), 2012. 5(2): p. 309-19.
- Thanos, J., et al., Adolescent dietary phytoestrogen intake and breast cancer risk (Canada). Cancer Causes Control, 2006. 17(10): p. 1253-61.
- Touillaud, M., et al., Use of dietary supplements containing soy isoflavones and breast cancer risk among women aged >50 y: a prospective study. Am J Clin Nutr, 2019. 109(3): p. 597-605.
- Constantine, G.D., et al., Increased Incidence of Endometrial Cancer Following the Women's Health Initiative: An Assessment of Risk Factors. J Womens Health (Larchmt), 2019. 28(2): p. 237-243.
- Alekel, D.L., et al., The soy isoflavones for reducing bone loss (SIRBL) study: a 3-y randomized controlled trial in postmenopausal women. Am J Clin Nutr, 2010. 91(1): p. 218-30.
- Unfer, V., et al., Endometrial effects of long-term treatment with phytoestrogens: a randomized, double-blind, placebo-controlled study. Fertil Steril, 2004. 82(1): p. 145-8, quiz 265.
- Tai, T.Y., et al., The effect of soy isoflavone on bone mineral density in postmenopausal Taiwanese women with bone loss: a 2-year randomized double-blind placebo-controlled study. Osteoporos Int, 2012. 23(5): p. 1571-80.
- Quaas, A.M., et al., Effect of isoflavone soy protein supplementation on endometrial thickness, hyperplasia, and endometrial cancer risk in postmenopausal women: a randomized controlled trial. Menopause, 2013. 20(8): p. 840-4.
- Jacobsen, B.K., et al., Soy isoflavone intake and the likelihood of ever becoming a mother: the Adventist Health Study-2. Int J Womens Health, 2014. 6: p. 377-84.
- Andrews, M.A., et al., Dietary factors and luteal phase deficiency in healthy eumenorrheic women. Hum Reprod, 2015. 30(8): p. 1942-51.
- Wiseman, H., et al., Isoflavone aglycon and glucoconjugate content of high- and low-soy U.K. foods used in nutritional studies. J Agric Food Chem, 2002. 50(6): p. 1404-10.
- Setchell, K.D., et al., Bioavailability of pure isoflavones in healthy humans and analysis of commercial soy isoflavone supplements. J Nutr, 2001. 131(4 Suppl): p. 1362S-75S.
- Ju, Y.H., et al., Effects of dietary daidzein and its metabolite, equol, at physiological concentrations on the growth of estrogen-dependent human breast cancer (MCF-7) tumors implanted in ovariectomized athymic mice. Carcinogenesis, 2006. 27(4): p. 856-63.

Reviewer 2 Report
The study is entitled: “Isoflavone consumption and risk of breast cancer: an up-2 dated systematic review with meta-analysis of observational studies and is proposed by Yang J, Shen H, Mi M, and Qin Y.
The study is interesting especially because it gives some clues about a dose response effect of isoflavone intake and the risk of breast cancers.
Abstract: The abstract does not fully reflect the results of the study and should be re-written. It should mention that intake doses corresponding to Western exposure i.e. <10 mg/day have no significant effects on the risk of breast cancer. Only Asian studies overall indicated a reduce risk of breast cancer associated to Soy intake. It should also mention that neither menopausal status nor ER status have a significant incidence on the link between soy intake and the risk of breast cancer. It should finally mention that the inverse link is significant in case-control studies and not in population studies.
Line 47. In the introduction, one of the most probable explanation for soy preventive effect on breast cancer is a inhibition of the cancer initiation. This should occur at early age while the breast cells are still in good shape. This means that such an effect most probably requires an isoflavone exposure during adolescence or in young women. When the cancer initiation is prevented, it results in an overall reduction of breast cancer incidence. This hypothesis which cannot be rejected by the present study should be mentioned in the introduction.
As a consequence, the most relevant criteria to analyse in meta-analysis is a sufficient isoflavone exposure at adolescence and during youth. Such an exposure is likely to occur in Asian people rather than in Western women.
Because at high doses isoflavones in human can be deleterious to reproduction, a dose-response assessment is highly relevant to be able to determine a dose range of exposure that could be beneficial for breast cancers without being detrimental to reproduction. Considering some recent studies, a dose below 50 mg/day seems not to induce reproductive impairment (https://doi.org/10.2147/IJWH.S57137 ; https://doi.org/10.1093/humrep/dev133 ; )
Line 75. In Western countries, the soy intake is generally underestimated since soy being used as an ingredient in many transformed dishes, isoflavones are consumed by a wide proportion of the population. In some cases this consumption can be in the range of 10 mg/days.
Line 139. As mentioned previously the starting time for isoflavone intake (childness, adolescence, adulthood, menopause) should have been analysed since it seems to be very important.
Line 271. Could you please mention a threshold i.e. the intake for which the reduction of breast cancer begins to be significant?
Line 348. It appears from the data obtained that the effect of an early exposure that would occur early enough to prevent the initiation phase of breast cancer cannot be excluded and this hypothesis which is supported by data obtained on Asian cohorts compared to cohorts of women who started taking isoflavones at adulthood (https://doi.org/10.1158/1940-6207.CAPR-11-0251) or on animal studies (https://doi.org/10.1093/carcin/16.11.2833) should be mentioned in the text.
The heterogeneity may also be explained by uncertainties in the soy intake assessment. Soy is more often used as an ingredient in transformed food nowadays than in the 2000’s. This ingredient significantly increase the exposure to isoflavones even in “non soy eaters”. Such uncertainty may explain the increase of heterogeneity in recent studies.
Line 375. there is something missing.
Author Response
Response to Reviewer 2 Comments
The study is entitled: “Isoflavone consumption and risk of breast cancer: an up-2 dated systematic review with meta-analysis of observational studies and is proposed by Yang J, Shen H, Mi M, and Qin Y.
The study is interesting especially because it gives some clues about a dose response effect of isoflavone intake and the risk of breast cancers.
Point 1: Abstract: The abstract does not fully reflect the results of the study and should be re-written. It should mention that intake doses corresponding to Western exposure i.e. <10 mg/day have no significant effects on the risk of breast cancer. Only Asian studies overall indicated a reduce risk of breast cancer associated to Soy intake. It should also mention that neither menopausal status nor ER status have a significant incidence on the link between soy intake and the risk of breast cancer. It should finally mention that the inverse link is significant in case-control studies and not in population studies.
Response 1: Thanks for your diligent suggestion. We have re-written the results of the abstract and demonstrated the points mentioned above. Please refer to line 22-26 of the revised manuscripts. The contents are as follows.
“Subgroup analysis further showed that neither menopausal status nor ER status has a significant influence on the association between isoflavone intake and breast cancer risk, while the isoflavone intake doses and study design does. When isoflavones exposure was less than 10 mg/day, no effects on breast cancer risk were detected. The inverse association is significant in case-control studies but not in cohort studies.”
Point 2: Line 47. In the introduction, one of the most probable explanation for soy preventive effect on breast cancer is a inhibition of the cancer initiation. This should occur at early age while the breast cells are still in good shape. This means that such an effect most probably requires an isoflavone exposure during adolescence or in young women. When the cancer initiation is prevented, it results in an overall reduction of breast cancer incidence. This hypothesis which cannot be rejected by the present study should be mentioned in the introduction. As a consequence, the most relevant criteria to analyse in meta-analysis is a sufficient isoflavone exposure at adolescence and during youth. Such an exposure is likely to occur in Asian people rather than in Western women.
Response 2: Thanks for your valuable comments which did a great favor in improving the assay writing. We have added the information of the hypothesis that the preventive effect of isoflavone on breast cancer is an inhibition of the cancer initiation, which is closely related to exposure period in lifetime. Since the isoflavone exposure period is important to the preventive effects based on the hypothesis but number of studies stratified by isoflavone exposure period is limited, our study only included the adult female data as described in the methods. However, the studies regarding to isoflavone exposure period have been comprehensively discussed in our revised manuscript line 356-374. The contents are shown below.
“One interesting hypothesis of the preventive effects of isoflavone on breast cancer is its inhibition of cancer initiation, which occurs at an early age when the cells are in good shape[1, 2]. It means that the starting exposure time of isoflavone is important for the prevention of breast cancer. However, a limited number of epidemical studies have been published that support the hypothesis [3-5]. A case-control study in Asian American females indicated that the high consumption of isoflavone during adolescence was related to the reduced risk of breast cancer, even though the consumption was low during adult life[3]. Thanos, et al proposed that higher exposure to isoflavones during adolescence exhibited a decreased risk of breast cancer among non-Asian women[5], and the subsequent study of this team showed that the decreased risk may be relative to the ER and PR status[4]. Further, A recent large prospective cohort study which included more than 11,000 women aged above 50 reported that isoflavones supplements had a significant positive association with ER-negative breast cancer although a significant inverse association with ER-positive breast cancer[6], which is inconsistent with our meta-analysis results that have included all adult female data, further suggesting that taking in isoflavone at an early age might benefit a lot to breast cancer.”
Point 3: Because at high doses isoflavones in human can be deleterious to reproduction, a dose-response assessment is highly relevant to be able to determine a dose range of exposure that could be beneficial for breast cancers without being detrimental to reproduction. Considering some recent studies, a dose below 50 mg/day seems not to induce reproductive impairment (https://doi.org/10.2147/IJWH.S57137; https://doi.org/10.1093/humrep/dev133)
Response 3: We agree that the safety of isoflavone is of great importance in the present study since we concluded a dose-dependently inverse association of isoflavone and breast cancer. The relative information has been added and discussed in line 442-461. The contents of are as follows.
“Although the association between isoflavone and breast cancer is dose-dependently inverse, the potentially detrimental effects of isoflavone at very high doses cannot be neglected. The leading concern is the potential detrimental effects on postmenopausal women since high concentrations of estrogen could lead to endometrial carcinoma in aged women[7]. One RCT study reported no significant adverse effects were observed in 224 United States postmenopausal women supplemented with soy isoflavone 120 mg/day for 3 years[8]. Another RCT study reported 6 women developed endometrial hyperplasia but no endometrial cancer in 319 Italian post-menopausal women supplemented with 150 mg/day soy isoflavones for 5 years[9]. More recently, no endometrial thickness hyperplasia or histopathological changes were observed in 399 postmenopausal women in Taiwan who were supplemented with 300 mg/day soy isoflavones for 2 years[10] and 350 postmenopausal women in the United States supplemented with 154 mg/day soy isoflavones for three years[11]. However, other potential hazards to the reproduction of high dose of isoflavone should be taken into consideration. Some recent observational studies report the potential detrimental effects on the reproduction of isoflavone with high doses in western women [12, 13], which should also be taken into consideration. Anyway, up-to-date evidence support that a dose below 50 mg/day may not induce reproductive impairment.”
Point 4: Line 75. In Western countries, the soy intake is generally underestimated since soy being used as an ingredient in many transformed dishes, isoflavones are consumed by a wide proportion of the population. In some cases, this consumption can be in the range of 10 mg/days.
Response 4: Thanks for your reminding of the potential underestimation of the isoflavone intake doses. The potential bias has been added to the limitation of our work in line 470-473. The contents are as follows.
“Secondly, the isoflavone intake might be underestimated in Western countries, since soy is used as an ingredient in many transformed dishes, which might lead to a bias in the subgroup analysis stratified by intake dose.”
Point 5: Line 139. As mentioned previously the starting time for isoflavone intake (childness, adolescence, adulthood, menopause) should have been analysed since it seems to be very important.
Response 5: We agree to your point that the starting time for isoflavone intake might be an important factor in the preventive effect of isoflavones on breast cancer. However, according to our inclusion and exclusion criteria as demonstration in line 79-95, only 3 studies[3-5] consist of data stratified by the isoflavone exposure time (adult vs. adolescent). Some studies involving isoflavone exposure time and the risk of breast cancer only reported soy food intake without estimating isoflavone intake doses[17, 18], thus being not included in our current analysis. Since we believe that the exposure time might attribute a lot in the preventive effect of isoflavone on breast cancer, our present study just included the adult data of the 3 above studies to calculate the summary OR to reduce the heterogeneity. Therefore, we consider that the subgroup analysis stratified by the starting time for isoflavone intake is inappropriate in the present study. Thank you for the comment and relative points have been added to the discussion about isoflavone starting exposure time as demonstrated in Response 2. (Line 356-374)
Point 6: Line 271. Could you please mention a threshold i.e. the intake for which the reduction of breast cancer begins to be significant?
Response 6: Thanks for the comments. We have added the descriptions of the results of dose-response meta-analysis in line 296-298 and 308-310 and mentioned the threshold as referred to the literature[19]. The contents as shown below.
“It can be observed that the risk of breast cancer significantly reduces while the isoflavone intake is around 15 mg/day, by the REMR method.” And “The descent of breast cancer risk is slightly decelerated when the isoflavone intake is >20mg/day, but no significant slowdown has been observed, in both of the methods.”
Point 7: Line 348. It appears from the data obtained that the effect of an early exposure that would occur early enough to prevent the initiation phase of breast cancer cannot be excluded and this hypothesis which is supported by data obtained on Asian cohorts compared to cohorts of women who started taking isoflavones at adulthood (https://doi.org/10.1158/1940-6207.CAPR-11-0251) or on animal studies (https://doi.org/10.1093/carcin/16.11.2833) should be mentioned in the text.
Response 7: Thanks for your comments. We have revised the manuscript and mentioned the two studies for further discussion on the isoflavone exposure time as demonstrated in Response 2 (line 356-374).
Point 8: The heterogeneity may also be explained by uncertainties in the soy intake assessment. Soy is more often used as an ingredient in transformed food nowadays than in the 2000’s. This ingredient significantly increase the exposure to isoflavones even in “non soy eaters”. Such uncertainty may explain the increase of heterogeneity in recent studies.
Response 8: Thank you for the helpful supplementary explanation on the heterogeneity. We have added this point in the discussion of heterogeneity among studies about publication time, line 379-385. The contents as shown below.
“One explanation could be the uncertainties in the isoflavone intake assessment. Soy, an isoflavone-rich food, is more often used as an ingredient in transformed food nowadays than in earlier times [20, 21]. This ingredient significantly increases the exposure to isoflavones even in “non-soy eaters”, which could probably underestimate the isoflavone intake dose, thus increasing the heterogeneity in recent studies.”
Point 9: Line 375. there is something missing.
Response 9: We have invited a colleague who is fluent in English to check and revise the manuscript all through to improve English writing. Thank you for your comments.
References:
- Lamartiniere, C.A., et al., Genistein suppresses mammary cancer in rats. Carcinogenesis, 1995. 16(11): p. 2833-40.
- Khan, S.A., et al., Soy isoflavone supplementation for breast cancer risk reduction: a randomized phase II trial. Cancer Prev Res (Phila), 2012. 5(2): p. 309-19.
- Wu, A.H., et al., Adolescent and adult soy intake and risk of breast cancer in Asian-Americans. Carcinogenesis, 2002. 23(9): p. 1491-6.
- Anderson, L.N., et al., Phytoestrogen intake from foods, during adolescence and adulthood, and risk of breast cancer by estrogen and progesterone receptor tumor subgroup among Ontario women. Int J Cancer, 2013. 132(7): p. 1683-92.
- Thanos, J., et al., Adolescent dietary phytoestrogen intake and breast cancer risk (Canada). Cancer Causes Control, 2006. 17(10): p. 1253-61.
- Touillaud, M., et al., Use of dietary supplements containing soy isoflavones and breast cancer risk among women aged >50 y: a prospective study. Am J Clin Nutr, 2019. 109(3): p. 597-605.
- Constantine, G.D., et al., Increased Incidence of Endometrial Cancer Following the Women's Health Initiative: An Assessment of Risk Factors. J Womens Health (Larchmt), 2019. 28(2): p. 237-243.
- Alekel, D.L., et al., The soy isoflavones for reducing bone loss (SIRBL) study: a 3-y randomized controlled trial in postmenopausal women. Am J Clin Nutr, 2010. 91(1): p. 218-30.
- Unfer, V., et al., Endometrial effects of long-term treatment with phytoestrogens: a randomized, double-blind, placebo-controlled study. Fertil Steril, 2004. 82(1): p. 145-8, quiz 265.
- Tai, T.Y., et al., The effect of soy isoflavone on bone mineral density in postmenopausal Taiwanese women with bone loss: a 2-year randomized double-blind placebo-controlled study. Osteoporos Int, 2012. 23(5): p. 1571-80.
- Quaas, A.M., et al., Effect of isoflavone soy protein supplementation on endometrial thickness, hyperplasia, and endometrial cancer risk in postmenopausal women: a randomized controlled trial. Menopause, 2013. 20(8): p. 840-4.
- Jacobsen, B.K., et al., Soy isoflavone intake and the likelihood of ever becoming a mother: the Adventist Health Study-2. Int J Womens Health, 2014. 6: p. 377-84.
- Andrews, M.A., et al., Dietary factors and luteal phase deficiency in healthy eumenorrheic women. Hum Reprod, 2015. 30(8): p. 1942-51.
- Wiseman, H., et al., Isoflavone aglycon and glucoconjugate content of high- and low-soy U.K. foods used in nutritional studies. J Agric Food Chem, 2002. 50(6): p. 1404-10.
- Setchell, K.D., et al., Bioavailability of pure isoflavones in healthy humans and analysis of commercial soy isoflavone supplements. J Nutr, 2001. 131(4 Suppl): p. 1362S-75S.
- Ju, Y.H., et al., Effects of dietary daidzein and its metabolite, equol, at physiological concentrations on the growth of estrogen-dependent human breast cancer (MCF-7) tumors implanted in ovariectomized athymic mice. Carcinogenesis, 2006. 27(4): p. 856-63.
- Shu XO, J.F., Dai Q, et al., Soyfood intake during adolescence and subsequent risk of breast cancer among Chinese women. Cancer epidemiology, biomarkers & prevention, 2001. 10(5): p. 483-488.
- Korde LA, W.A., Fears T, et al., Childhood soy intake and breast cancer risk in Asian American women. Cancer Epidemiol Biomarkers Prev, 2009. 18(4): p. 1050-1059.
- Ha, A., et al., Degree of Myopia and Glaucoma Risk: A Dose-Response Meta-analysis. Am J Ophthalmol, 2022. 236: p. 107-119.
- Lee, A., et al., New Evaluation of Isoflavone Exposure in the French Population. Nutrients, 2019. 11(10).
- Villares, A., et al., Content and Profile of Isoflavones in Soy-Based Foods as a Function of the Production Process. Food & Bioprocess Technology, 2011. 4(1): p. 27-38.

Round 2
Reviewer 1 Report
No further comments.